# Exploring the Dynamic Nexus between Cross-Border Dollar Claims and Global Economic Growth

**Constantinos Alexiou** [1], **Sofoklis Vogiazas** [2,*] **and Alex Benbow** [1]

1   Cranfield School of Management, Cranfield University, Cranfield MK43 0AL, UK;
    constantinos.alexiou@cranfield.ac.uk (C.A.); alex.benbow@cranfield.ac.uk (A.B.)
2   Black Sea Trade and Development Bank, Financial Analysis & Risk Division & Benbow,
    54624 Thessaloniki, Greece
*   Correspondence: svogiazas@bstdb.org

**Abstract:** This paper addresses the role of the U.S. dollar in fostering global economic growth during the post-war period. The existing literature lacks a comprehensive understanding of the true implications of the U.S. dollar's status as a reserve currency and a dearth of studies examining its impact. In this study, we explore the dynamic long-run and short-run relationships between cross-border U.S. dollar claims, global GDP, and global trade while gauging the impact of the Global Financial Crisis (GFC) and the COVID-19 pandemic. In doing so, we use ARDL methodology for a data set that spans the period of 1980 to 2022. The estimation results reveal a robust long-run relationship between U.S. dollar claims, global GDP and global trade and no clear evidence of asymmetric effects. Our findings are of great significance for monetary authorities, emphasising the need for a nuanced understanding of the implications of the U.S. dollar's conducive role in shaping global economic dynamics and fostering growth.

**Keywords:** eurodollar; cross-border dollar claims; economic growth; global trade

**JEL Classification:** C22; F31; F40

## 1. Introduction

The term "Eurodollar" first appeared in the public realm in 1960 when William Clarke gave name to a fledgling market (Clarke 1960). Nowadays, the terms "Eurodollar", "Eurodollar system", "ledger money", "shadow money" and "ghost money" are all used interchangeably to describe U.S. dollars that exist offshore, that is, outside the U.S. domestic currency system and out of sight of its primary regulator, the Federal Reserve.

The precise origin of the Eurodollar market remains a topic of debate. Some authors trace it to post-war U.S. trade deficits, political or regulatory factors, the petrodollar and oligarchs attempting to create haven asset pools outside the United States. Ultimately, as its name suggests, it found its footing in Europe, primarily in the City of London. However, other financial centres worldwide are also cited as key contributors to its proliferation (Friedman 1971, 1993). Nowadays, the Eurodollar market exists everywhere and touches practically every part of the global economy.

Despite the U.S. economy's declining share of global GDP, its currency remains dominant as the medium of exchange in international trade, debt issuance and financial transactions. Thanks to its liquidity, almost 90% of foreign exchange transactions worldwide involve the U.S. dollar. At the heart of this dollar internationalization is an "offshore" banking and currency system referred to as the Eurodollar system. The Eurodollar market is about short-term deposits denominated in U.S. dollars at banks outside U.S. territory. The growth of the Eurodollar market has been linked to the Bretton Woods agreement and the emergence of capital control measures in many nations.

The expansion of the Eurodollar system ran parallel with post-war economic growth. The abundance of dollars globally was likely a significant contributing factor to growth. As money primarily functions as a medium of exchange and does not contribute to economic growth, it is reasonable to assert that the availability of dollars did not act as a limiting factor for global growth and globalization during the post-war period, at least until 2007. Whilst a review of the academic literature would suggest widespread recognition that the U.S. dollar has played a significant role in post-war growth and globalization, there is scarce empirical analysis in this area. We could not find any direct empirical evidence on the relationship between Eurodollar deposit growth, global economic growth and global trade. Exploring the strength of the relationship becomes more challenging, especially in the aftermath of the two most recent critical events, the Global Financial Crisis (GFC) and COVID-19.

During the GFC in 2007–2008, there was a significant impact on cross-border financial flows and claims. Financial institutions faced liquidity and solvency issues, leading to a sharp reduction in cross-border lending. Many banks faced a decline in their capital positions, and the interbank lending market experienced disruptions. Cross-border dollar claims, particularly loans and financial instruments were affected as financial institutions became risk averse. Central banks and international organisations took measures to stabilise financial markets and prevent a systemic collapse (Cavallino and De Fiore 2020). However, the crisis highlighted the interconnectedness of global financial markets and the potential for contagion.

Furthermore, the COVID-19 pandemic had widespread and profound effects on the global economy, financial markets and cross-border financial activities (Adrian et al. 2022). The pandemic led to disruptions in international trade, supply chains, and economic activities, affecting various sectors. Governments worldwide implemented measures such as lockdowns and travel restrictions to contain the spread of the virus, which had cascading effects on financial markets. During the early stages of the pandemic, there was a flight to safety, with investors seeking refuge in traditional safe-haven assets such as the U.S. dollar. The latter contributed to an appreciation of the dollar and affected cross-border funding conditions. The economic downturn and the pandemic-related uncertainties also led to increased credit risk and financial market volatility. Globally, central banks responded with monetary and fiscal measures to support their economies and stabilise financial markets (Cavallino and De Fiore 2020). Regarding cross-border dollar claims, the impact varied depending on the exposure of financial institutions and the sectors they were involved in. Some countries and businesses faced challenges in servicing their dollar-denominated debt, particularly if the pandemic severely hit their economies.

Given the above, this paper aims for the first time to empirically estimate the relationship between the buildup of offshore dollar deposits (Eurodollars) or USD cross-border claims, global economic growth and global trade, while controlling for the impact of the Global Financial Crisis (GFC) and the COVID-19 pandemic. In doing so, by employing the ARDL and NARDL methodologies for a data set that spans the period of 1980 to 2022, we explore the short and long-term dynamics and the causal dimension of the underlying factors. The estimation results reveal a robust long-run relationship between U.S. dollar claims, global GDP and global exports, whilst no clear evidence of asymmetric effects is established. Further, we find that Eurodollar Deposit Growth (or U.S. claims) does not drive the short-term relationship, whilst the impact of the Global Financial Crisis (GFC) and the COVID-19 pandemic is found to be insignificant. In this context, we argue that economic growth has led to the accumulation of U.S. dollar deposits for several other reasons (trade competitiveness, investment flows and balance of payments imbalances).

The remainder of this paper is organised as follows. Section 2 provides a contextual overview of cross-border claims regarding the Eurodollar, whilst Section 3 reviews the theoretical background that relates to the research area. Section 4 presents the data selection and sets out the methodology applied, and Section 5 reports and discusses the results

of the empirical analysis. Finally, Section 5 provides the concluding remarks, including implications for monetary policy and proposals for future research.

## 2. Contextual Overview

The process of Eurodollar creation is illustrated by Friedman (1971), who described it as a process of money creation by 'a bookkeeper's pen'. Simply put, it is the process of fractional reserve banking largely void of any regulatory reserve requirements. This, in theory, creates an offshore pool of largely unrestricted money supply, with the primary constraint being the cash-flow limitations of banks rather than regulatory boundaries. However, even this cash-flow constraint has been stretched to its limits, with banks creating resourceful new methods to fund their short-term cash-flow requirements as cheaply as possible.

Although the term Eurodollar describes only offshore dollar deposits, offshore 'ledger money' exists for other reserve currencies[1], but to a much smaller degree. It is not easy to demonstrate directly the impact of the Eurodollar on either financial markets or the economy, as monetary authorities do not publish the Eurodollar volume. What astronomists call 'dark matter' provides a fitting metaphor for Eurodollars that exist mainly in the shadows. Whilst scientists cannot directly observe dark matter or dark energy, they can detect its existence by measuring its effects on visible matter in the universe. Restrepo-Echavarría and Grittayaphong (2022) provide a detailed account of the growth of the Eurodollar market.

The shortage of Eurodollars was evident in the actions of the Federal Reserve during the Global Financial Crisis (GFC). For example, of the USD 20 billion Term Auction Facility (TAF) offered in December 2007, over USD 14.3 billion (or 72%) was taken up by U.S. subsidiaries of European banks. They were not global megabanks either; some of the largest bidders were banks like Landesbank Baden Wuerttemb and Bayerische Landesbank, which were hardly household names (Snider 2019). The infatuation with the role of subprime mortgages in the crisis has prevented economists from seeing the bigger picture. Subprime lending may have been the catalyst, but the dollar played a critical role.

Attempts to accurately quantify the Eurodollar system are likely to be in vain. Regulators have not required banks to report the transactions they undertake to capture the full scope of the Eurodollar system, which would possibly require a globally coordinated effort. Therefore, the data necessary to measure the size and growth of the Eurodollar system are not readily available. Through its FR 2420 report, the Federal Reserve does require daily reporting of Eurodollar transactions conducted by U.S. domestic banks and U.S. branches of foreign banking organisations. It essentially captures the portion of the Eurodollar market that operates within the U.S. Although it is noted that this overnight brokered Eurodollar market is around three to four times larger than the overnight Fed Funds (Cipriani and Gouny 2015), it can still only be assumed that this captures a thin slice of the overall Eurodollar market. At various points, researchers and monetary authorities have tried to estimate the size of the Eurodollar system. However, this task has become increasingly challenging, as the system has grown in size and complexity.

In the extant literature, 'dollarization' refers to the use by a country's residents of assets (or liabilities) denominated in another country's currency. Dollarization may be partial depending on whether a domestic currency circulates in parallel with the foreign currency. However, in some cases, the term has also been used to describe what we will distinguish as 'unofficial dollarization', where this process occurs involuntarily. The term 'currency internationalization' describes using a select few domestic currencies, which are increasingly used to facilitate global trade and capital flows. While it is generally agreed that foreign banks face a trade-off between the typically lower interest rates available on dollar deposit liabilities and the increased default risk arising on balance sheet currency mismatches, there is a broad range of views on the incentives to borrow in dollars. Broda and Yeyati (2006) suggest dollarization can take several forms, including foreign borrowing and deposit dollarization. Yet, the balance of the literature is weighted towards addressing the tendency of foreign banks to borrow in dollars.

### 3. Theoretical Background

The relationship between the U.S. dollar and global economic growth has been a topic of significant interest and debate among economists, policymakers and researchers. However, there is a lack of academic literature examining the role of the U.S. dollar as the global reserve currency on global economic growth. As such, our effort to provide a comprehensive empirical review in the area is hampered by the lack of studies. We have therefore endeavoured to provide a brief literature review of the most relevant studies that investigate the impact of dollar claims on various dimensions of the economy, including the role of the U.S. dollar as a global reserve currency, its influence on trade, and its implications for emerging market economies.

The U.S. dollar is often considered a haven, especially during periods of global economic uncertainty. Eichengreen and Flandreau (2012) discuss the role of the dollar as a safe haven and its implications for financial markets and economic growth and, in this context, point out that changes in global economic conditions that affect the demand for the U.S. dollar as a safe asset are crucial for anticipating its impact on the broader economy.

The dominance of the U.S. dollar as the world's primary reserve currency has been a defining feature of the international monetary system. Obstfeld and Rogoff (2000) and Eichengreen (2011) have explored the benefits and challenges associated with the dollar's status as a global reserve currency. They argue that the use of the dollar in international trade and finance provides the U.S. with significant advantages but also creates vulnerabilities for the global economy.

Furthermore, on the impact of dollar claims on global trade patterns and exchange rates, Gopinath and Stein (2018) examine the dollar's role in facilitating international trade, emphasising its importance in pricing and invoicing. Additionally, Goldberg and Tille (2008) suggest that "the U.S. dollar appears to be important in the invoicing of world trade both because the U.S. is an important consumer and producer in world markets, and because of its use in invoicing the many products that are traded via organised exchanges or using reference pricing" (p. 185).

Many emerging market economies face challenges related to dollar claims, such as high levels of dollar-denominated debt and the risk of currency crises. Jeanne and Rancière (2006) provide insights into dollarization in these economies, examining its impact on economic stability and growth. The authors highlight the potential benefits and risks associated with adopting the U.S. dollar to reduce transaction costs and gain access to international capital markets. In another study, Frankel and Saravelos (2012) analyse the dollar's role in predicting financial crises, emphasising its significance as a leading indicator. They argue that understanding the dynamics of financial crises and the role of the U.S. dollar is crucial for policymakers seeking to enhance global economic stability.

The literature on financial globalization explores how the growth of international financial markets, often dominated by the U.S. dollar, influences economic development. Reinhart and Reinhart (2008) analyse the relationship between financial globalization, dollar claims, and economic crises, emphasising the role of currency mismatches and the potential for contagion. Their work contributes to understanding the complex interplay between global financial integration and economic growth.

From a U.S. perspective, the dollar's hegemony has enabled the country to run perpetual trade deficits for decades (Bernanke 2005). Heightened demand for dollars increases the dollar's exchange value and reduces the competitiveness of U.S. exports while reducing the relative cost of imports. Likewise, insatiable global demand for dollar assets, particularly U.S. Treasury Bills, has permitted the U.S. Federal Government to run sustained budget deficits without incurring punitive rates (Bernanke 2010).

The supply of Eurodollars is discretionary and determined by the global banking system, influenced by factors largely outside the control of the individual countries it impacts. Small emerging economies, in particular, have had little choice but to accept the encroachment of the dollar into their economic system. In the post-war period, this granted them access to global trade and cheap dollar credit, exposing them to significant currency

risk. Bordo et al. (2009) empirically demonstrate that increasing foreign debt elevates the probability of both currency and debt crises. The authors also find that these crises lead to permanent losses in economic output. The Mexican 'tequila crisis' (1994), the Russian ruble crisis (1998) and the 'Asian Flu' can all be largely attributed to significant U.S. dollar liabilities (Eichengreen and Hausmann 1999).

The existence of the Eurodollar presents some challenges for central banks, particularly in the context of the monetarist doctrine. In the monetarist view, control over the supply of money is seen as the most effective means to control the price level (McClam 1980). The emergence of a fractions-on-fractions reserve-based system, and subsequently a reserveless system, would undoubtedly be an affront to monetarists. Eurocurrencies are both challenging to locate and to quantify.

Frydl (1982) admits that the Eurodollar poses a conundrum for the Federal Reserve, both in terms of setting policy based on monetary aggregates, which are exclusive of Eurodollars, as well as from a lender of last resort perspective, should the Fed be required to lend to non-U.S. banks. Interestingly, Frydl (1982) puts the Eurodollar's continued expansion down to its past stability, i.e., survivorship bias. Then, it is worth exploring whether this survivorship bias attached to Eurodollar growth could be derailed by an exogenous event such as the GFC.

Bernanke (2010) alludes to a shadow banking system offshore dollar reserve balances (Bernanke 2005) and even identifies that during the crisis, "heavy foreign demand for dollar funding began to disrupt money markets and squeeze credit availability in the United States" (Bernanke 2010). Snider (2013) demonstrates that after removing the reserve requirements of Eurodollar deposits in 1990, the U.S. financial system quickly moved from being a net lender into the Eurodollar system to being a net borrower. Snider (2013) also shows that what occurred during the GFC was not a supply problem but rather one of flow caused by fragmentation in interbank wholesale markets. Immediately after the GFC, both U.S. domestic and foreign banks' participation in Eurodollar markets collapsed (Snider 2013).

While it is evident from the existing literature that researchers accept that the role of the U.S. dollar has been a significant factor in global growth, trade–capital flows, and financial instability, there is limited empirical research directed at studying these relationships. Furthermore, existing research focuses on the experience of an individual country or region. Yet, it is clear that the Eurodollar system is global, and we should expect its effects on economic growth and trade also to be global.

Given the above and in light of the scarce empirical evidence, we set out to explore the following research questions:

RQ1: Is there a short/long-run relationship between economic growth, global trade and rising offshore Eurodollar balances?

RQ2: Is there a causal relationship between rising offshore dollar balances (Eurodollar), global trade and economic growth?

## 4. Data and Methodology

### 4.1. Data Selection

The empirical analysis focuses on exploring the relationship between World Economic Growth, the Eurodollar Deposit Growth and World Trade, allowing for the effect of destabilising events such as the GFC and COVID-19 for the period of 1980–2022. Table 1 presents the variables used in our analysis, while Table 2 presents the descriptive statistics.

Eurodollar data or USD cross-border claims (also "USD claims") have been obtained from BIS locational banking statistics, specifically the 'A2 cross-border positions' data set, using cross-border claims denominated in U.S. dollars. These reflect the outstanding claims of internationally active banks located in reporting countries on counterparties residing in more than 200 countries and capture around 95% of all cross-border banking activity.

**Table 1.** Variables.

| Variable | Proxies |
|---|---|
| Eurodollar Deposit Growth | USD Cross-Border Claims |
| World Economic Growth | Global (World) GDP |
| World Trade | Global (World) Exports |
| Global Financial Crisis | Dummy in line with NBER |
| COVID-19 | Dummy in line with NBER |

Source: BIS, World Bank, IMF, NBER.

**Table 2.** Descriptive statistics.

| Statistic | Global GDP | Global Exports | USD Cross-Border Claims |
|---|---|---|---|
| Mean | 31.2407 | 29.5528 | 24.0250 |
| Median | 31.1528 | 29.5014 | 23.7889 |
| Standard Deviation | 0.6886 | 0.8766 | 1.1011 |
| Kurtosis | 1.8505 | 1.6393 | 2.0445 |
| Skewness | −0.2531 | −0.1883 | 0.3833 |
| Minimum | 30.0591 | 28.1154 | 22.2890 |
| Maximum | 32.2418 | 30.8270 | 26.0982 |
| Observations | 43 | 43 | 43 |

Note: data are expressed in logarithmic form.

Some points on the selection of this data series need to be addressed. Firstly, it should not be interpreted that these data include the entire Eurodollar market. However, the chosen data reflect a significant component of the Eurodollar system. Therefore, we can make inferences about the growth rate of the entire Eurodollar system. Also, the data set includes reporting by U.S. banks (i.e., claims of U.S. banks on overseas banks) and overseas bank claims on U.S. banks. Whilst claims on U.S. banks do not meet the strict definition of Eurodollars (i.e., offshore U.S. dollar deposits), this does reflect borrowing by U.S. banks within the Eurodollar market. On balance, it is viewed as a valid inclusion in the data set and has, therefore, not been adjusted for.

In Figure 1, we plot our key variables in logarithmic form. In the case of global GDP and global exports, we can easily visualise a stable and upward trend with potential breaks during the GFC and COVID-19 periods. However, this is not the case with U.S. cross-border claims, where the series' evolution is not even in the period under investigation. This is because these data are subject to frequent reviews and breaks in series that may arise from changes in the population of reporting institutions, including the addition of new reporting countries, changes in reporting practices, or methodological improvements (BIS 2021). This is simply an unavoidable limitation of these data and, therefore, should be accepted as such.

As noted, both the GFC and COVID-19 coincided with structural breaks in our time series, especially in global economic growth (global GDP) and global exports. We incorporate this information into our analysis using two dummy variables, GFC and COVID-19, in line with NBER's business cycle occurrence.

The economic output should follow suit with the expansion or contraction of the Eurodollar volume. It may also be the case that expanding or contracting GDP drives Eurodollar volume as lenders' decisions to extend credit (dollars or otherwise) are likely motivated by perceptions about economic conditions. Therefore, a positive relationship is expected between cross-border USD claims and global GDP, although the direction in which this relationship runs is unclear.

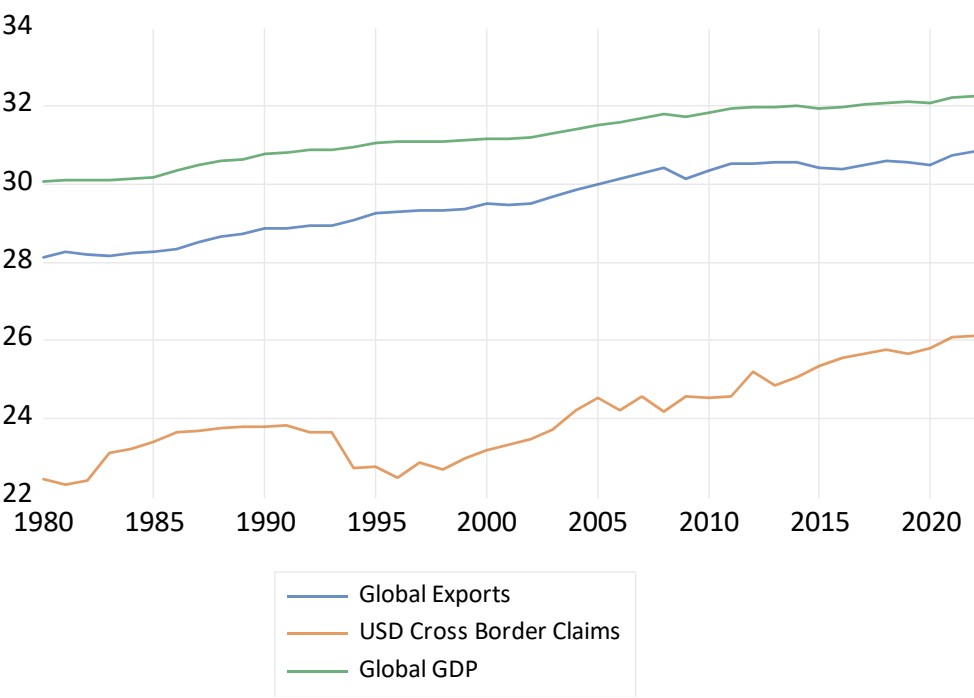

**Figure 1.** Plot of global GDP, global exports and USD cross-border claims.

Equally, given that the origins of the Eurodollar were principally about providing a means of settlement for global trade, the relationship between Eurodollar growth and global exports is expected to be positive. Without sufficient Eurodollars to 'grease the wheels' of trade, we would expect growth in gross global exports/imports to slow. Ceteris paribus, a country's exports should grow proportionally to its GDP; therefore, the relationship between GDP and global trade is also expected to be positive.

*4.2. Methodology*

For the purpose of our analysis and considering the lack of empirical studies in the research area, we use the ARDL framework as an adequate method of examining cointegrating relationships. Two seminal contributions from Pesaran and Shin (1998) and Pesaran et al. (2001) argue that ARDL models are especially advantageous in their ability to handle cointegration with inherent robustness to misspecification of integration orders of relevant variables. Furthermore, if a cointegrating relationship exists among the variables of interest, we can estimate the error correction term, which indicates the speed of convergence towards the long-run equilibrium. By using our terminology, the general ARDL (p, q1, q2) model is given by the following equation:

$$\psi(L)Global\ GDP_t = a_0 + a_1 t + \sum_{j=1}^{2} \beta_j(L)x_{j,t} + GFC + COVID\text{-}19 + \varepsilon_t$$

where p, q1 and q2 give the lag order for the variables used, $\varepsilon_t$ are the usual innovations, $a_0$ is a constant term and $a_1$ is the coefficient associated with the linear trend. Then, $L$ denotes the usual lag operators, along with the respective lag polynomials, and $x_1$ and $x_2$ are the two explanatory variables used in our study (global trade and Eurodollar claims). Then, the variables GFC and COVID-19 are modelled as exogenous variables.

It should be stressed that even though the specific methodology takes care of a mixed order of integration, i.e., $I(0)$ and $I(1)$, we have also checked for unit roots to ensure that none of the series is $I(2)$ (Alexiou and Vogiazas 2019). The unit root tests in Appendix A (Table A1) indicated that all other series are integrated of order 1, i.e., $I(1)$.

### 5. Empirical Results and Discussion

Following the model specification and estimation, we run the error diagnostic tests to ensure that residuals are serially uncorrelated and homoscedastic, while we also test the model's stability. The results are presented in Table A2 and Figure A1 in the Appendix A.

To test for the presence of cointegration, we perform the bounds test. The F-statistic value is 8.34, which is greater than the $I(1)$ critical value bound of 5.8 at the 1% significance level, indicating that we reject the null hypothesis that there is no equilibrating relationship. As outlined in Pesaran et al. (2001), the rejection of the t-bounds test in the secondary stage confirms the existence of a cointegrating relationship. Still, it does not preclude that it is degenerate. To rule out a degenerate cointegration, a joint test of parameter significance on all coefficients associated with distributed lag variables in levels ought to be inspected. With a *p*-value of 0.00, the Wald test rejects the null hypothesis that all tested coefficients are jointly zero and, by extension, confirms that the cointegrating relationship that will emerge is sensible and not degenerate. Thus, it makes sense to study the speed of adjustment equation presented in Table 3.

**Table 3.** Dynamic and long-run estimates (ARDL).

| | Short-run effects | | | |
|---|---|---|---|---|
| Variable | Coefficient | Std. Error | *t*-Statistic | Prob. |
| DLOG(USD_CLAIMS) | 0.0145 | 0.0123 | 1.1804 | 0.2456 |
| DLOG(USD_CLAIMS(−1)) | −0.0379 | 0.0133 | −2.8555 | 0.0071 |
| DLOG(GLOBAL_EXPORTS) | 0.4636 | 0.0350 | 13.23993 | 0.0000 |
| COVID-19 | −0.0248 | 0.0233 | −1.0621 | 0.2953 |
| GFC | 0.0265 | 0.0167 | 1.586266 | 0.1214 |
| COINTEQ | −0.1207 | 0.020011 | −6.03135 | 0.0000 |
| | Long-run effects | | | |
| LOG(USD_CLAIMS) | 0.2163 | 0.119 | 1.815 | 0.077 |
| LOG(GLOBAL_EXPORTS) | 0.5137 | 0.155 | 3.322 | 0.002 |
| C | 11.062 | 2.038 | 5.429 | 0.000 |
| R-squared | 0.8345 | | | |
| Adjusted R-squared | 0.812 | | | |
| S.E. of regression | 0.0225 | | | |
| Sum squared resid | 0.0183 | | | |
| Log likelihood | 102.932 | | | |
| F-statistic | 36.417 | | | |
| Prob(F-statistic) | 0.000 | | | |

As expected, the error correction term, here represented as COINTEQ, is negative with an associated coefficient estimate of −0.1207. This implies that about 12.1% of any movements into disequilibrium are corrected within one period. Moreover, given the very large t-statistic, namely −6.03, we can also conclude that the coefficient is highly significant.

Table 4 also presents the estimated coefficients and the respective t-statistics of the long-term cointegrating relationship. Clearly, both Eurodollar claims and global exports are significant determinants of global economic growth in the 10% and 1% levels of significance in the long run. At the same time, exogenous factors such as the GFC and COVID-19 cannot impact the established relationship, even in the short run.

These findings are in line with expectations and broadly consistent with other studies considering the dearth of studies in the area (see, for example, Kelly et al.'s (2013) study for Ireland, and Ganchev et al.'s (2014) study for Eastern European economies).

The classical ARDL framework assumes the long-run relationship is a symmetric linear combination of regressors. While this is a natural starting assumption, it does not match the approach of the economics literature to modelling nonlinearity. Therefore, following the ARDL estimation, we also applied the NARDL framework to investigate nonlinearity assumptions. The NARDL model is flexible enough to accommodate even

partial asymmetry. This manifests when variables enter asymmetrically either the adjusting or cointegrating dynamics. Table 4 reports the results of the symmetry tests.

**Table 4.** Symmetry tests for global exports and USD claims.

| Coefficient symmetry tests | | | |
|---|---|---|---|
| Null hypothesis: coefficient is symmetric | | | |
| Variable | Statistic | Value | Probability |
| Long-run | | | |
| GLOBAL_EXPORTS | F-statistic | 1.938387 | 0.174079 |
| | Chi-square | 1.938387 | 0.163844 |
| Short-run | | | |
| GLOBAL_EXPORTS | F-statistic | 1.040627 | 0.315832 |
| | Chi-square | 1.040627 | 0.307676 |
| Joint (Long-run and Short-run) | | | |
| GLOBAL_EXPORTS | F-statistic | 1.732139 | 0.194132 |
| | Chi-square | 3.464278 | 0.176906 |
| Variable | Statistic | Value | Probability |
| Long-run | | | |
| USD_CLAIMS | F-statistic | 0.285093 | 0.59719 |
| | Chi-square | 0.285093 | 0.593382 |
| Short-run | | | |
| USD_CLAIMS | F-statistic | 0.02023 | 0.887817 |
| | Chi-square | 0.02023 | 0.886896 |
| Joint (Long-run and Short-run) | | | |
| USD_CLAIMS | F-statistic | 0.164464 | 0.849084 |
| | Chi-square | 0.328927 | 0.848349 |

The tests for long- and short-run symmetry and the joint test for symmetry for both global exports and USD claims suggest that we cannot reject the null hypothesis of symmetry at all reasonable significance levels. For consistency reasons, we report the results from the NARDL approach in Tables A3 and A4 in Appendix A.

As a final step and in order to establish the directional relationship between Eurodollar growth (USD claims), global GDP and global trade, the Granger Causality testing was performed. As each of the selected variables is interlinked and co-related through multiple channels, it is critical to gain insight into the relationship's causal dimension. We report the results of the Granger Causality tests in Table 5.

**Table 5.** Pairwise Granger Causality tests.

| Null Hypothesis | F-Statistic | Prob |
|---|---|---|
| GLOBAL_GDP does not Granger Cause USD_CLAIMS | 0.2044 | 0.8156 |
| USD_CLAIMS does not Granger Cause GLOBAL_GDP | 1.4571 | 0.2463 |
| GLOBAL_GDP does not Granger Cause GLOBAL_EXPORTS | 3.23236 | 0.0512 |
| GLOBAL_EXPORTS does not Granger Cause GLOBAL_GDP | 1.93455 | 0.1592 |
| GLOBAL_EXPORTS does not Granger Cause USD_CLAIMS | 0.52330 | 0.5970 |
| USD_CLAIMS does not Granger Cause GLOBAL_EXPORTS | 0.87289 | 0.4264 |

The results in Table 5 indicate that in almost all cases, we cannot reject the null hypothesis of no Granger Causality between the variables of interest. The only case where we marginally reject the null hypothesis of no Granger Causality is between global GDP and global exports. However, it is essential to note that the Granger Causality does not provide insights into the true causal relationship between two variables, while it is mainly used for ascertaining short-run causality.

## 6. Concluding Remarks

This paper addresses the underappreciated role of the U.S. dollar in fostering global economic growth during the post-war period. The existing literature lacks a comprehensive understanding of the true implications of the U.S. dollar's status, and there is a notable absence of studies examining the impact of U.S. cross-border claims, jointly with global trade, on inducing economic growth in a worldwide context. The dynamic short-run relationships and long-run convergence between cross-border U.S. dollar claims, global GDP, and global exports are investigated, whilst at the same time the effect of the Global Financial Crisis (GFC) and the COVID-19 pandemic are evaluated.

The estimation results reveal a robust cointegrating relationship between offshore U.S. dollar claims, global GDP and global exports. At the same time, we do not find clear evidence of asymmetric effects deriving from U.S. dollar claims and global exports to global economic growth, whilst the GFC and COVID-19 dummies are found to be insignificant.

Our research findings hold immense importance for central banks and other monetary authorities. They underscore the necessity for a nuanced comprehension and deep appreciation of the far-reaching effects stemming from the U.S. dollar's pivotal role. This role extends beyond mere currency exchange, it actively shapes global economic dynamics and provides crucial support for growth on a global scale. Even during periods of instability and uncertainty, the U.S. dollar remains a stabilising force, fostering economic resilience worldwide.

The availability of Eurodollar data is a limitation of this study. As discussed, accurately measuring offshore dollar balances is not an attainable goal, as no authority publishes complete Eurodollar transactional data while the available series are subject to frequent reviews. Equally, whilst this paper aimed to demonstrate the relationship between Eurodollar growth and global economic growth, other critical relationships might be worth exploring, such as the relationship between Eurodollar historical contractions and financial instability.

**Author Contributions:** Conceptualization, A.B.; methodology, S.V. and C.A.; validation, S.V. and C.A. formal analysis, S.V. and C.A.; investigation, A.B., S.V. and C.A.; writing—original draft preparation, A.B.; writing—review and editing, C.A. and S.V.; supervision, C.A.; funding acquisition, S.V.. All authors have read and agreed to the published version of the manuscript.

**Funding:** This research received no external funding.

**Informed Consent Statement:** Not applicable.

**Data Availability Statement:** All data used in this study are publicly available data and can be retrieved from the sources identified in Table 1. No new data were created.

**Conflicts of Interest:** The authors declare no conflict of interest.

## Appendix A

**Table A1.** Unit Root tests.

| Augmented Dickey–Fuller Test | Level | | First Differences | |
|---|---|---|---|---|
| **Variable** | *t*-**Statistic** | **Prob** | *t*-**Statistic** | **Prob** |
| GLOBAL_GDP | −1.462639 | 0.5427 | −5.114969 | 0.0000 |
| GLOBAL_EXPORTS | −0.776608 | 0.8155 | −6.290875 | 0.0000 |
| USD_CLAIMS | −0.404279 | 0.8991 | −7.521403 | 0.0000 |

**Table A2.** Serial Correlation, Heteroskedasticity Tests.

| **Breusch–Godfrey Serial Correlation LM Test** | | | |
|---|---|---|---|
| F-statistic | 0.013696 | Prob. F(2,31) | 0.986403 |
| Obs × R-squared | 0.03708 | Prob. Chi-square(2) | 0.981631 |
| Heteroskedasticity Test: Breusch–Pagan–Godfrey | | | |
| F-statistic | 0.981008 | Prob. F(8,33) | 0.467931 |
| Obs*R-squared | 8.069386 | Prob. Chi-square(8) | 0.426722 |
| Scaled explained SS | 13.20082 | Prob. Chi-square(8) | 0.105124 |

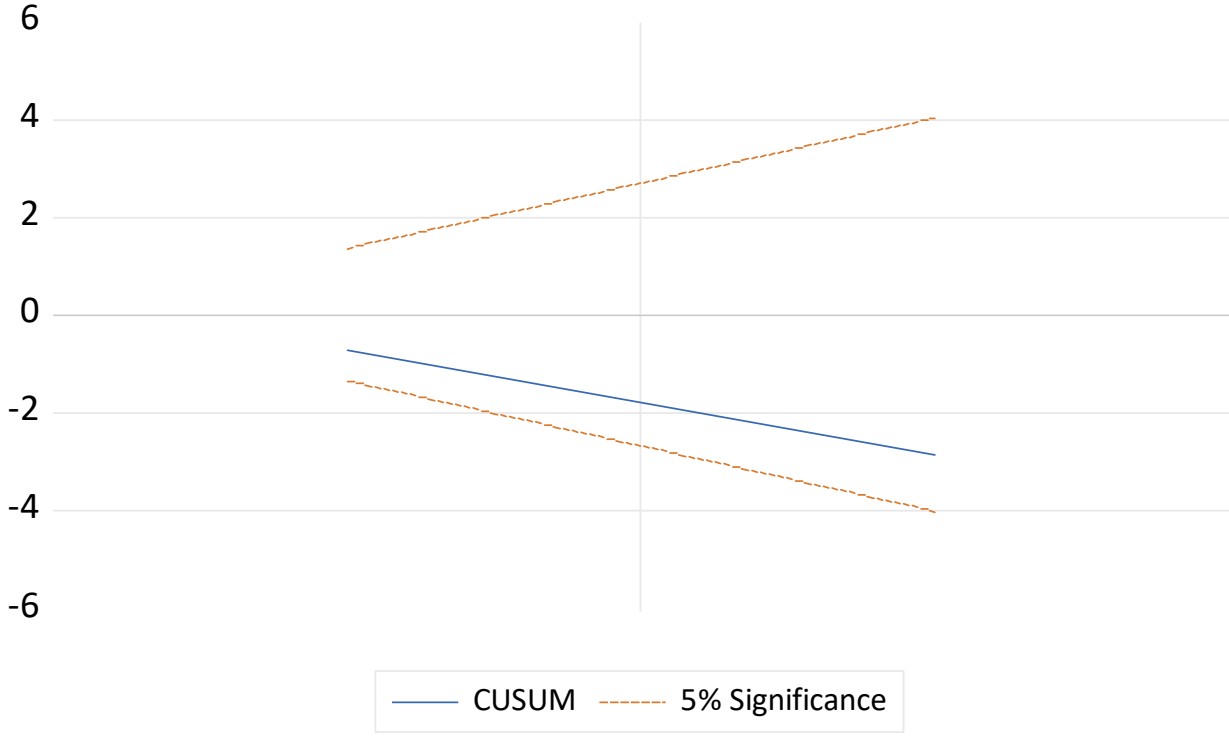

**Figure A1.** Parameter Instability Test (CUSUM).

**Table A3.** Dynamic and long-run estimates (NARDL)—global exports.

| Short-run effects | | | |
|---|---|---|---|
| Variable | Coefficient | Std. Error | t-Statistic | Prob. |
| DLOG(USD_CLAIMS) | 0.0141 | 0.0114 | 1.2370 | 0.2246 |
| DLOG(USD_CLAIMS(-1)) | −0.0292 | 0.0118 | −2.4806 | 0.0182 |
| @DCUMDP(LOG(GLOBAL_EXPORTS)) | 0.5011 | 0.0377 | 13.2941 | 0.0000 |
| @DCUMDN(LOG(GLOBAL_EXPORTS)) | 0.4092 | 0.0879 | 4.6541 | 0.0000 |
| COVID-19 | −0.0145 | 0.0209 | −0.6921 | 0.4936 |
| GFC | 0.0090 | 0.0186 | 0.4811 | 0.6336 |
| COINTEQ | −0.1249 | 0.0202 | −6.1989 | 0.0000 |
| Long-run effects | | | |
| LOG(USD_CLAIMS(−1)) | 0.2537 | 0.1542 | 1.6456 | 0.1083 |
| @CUMDP(LOG(GLOBAL_EXPORTS(−1))) | 0.5493 | 0.1198 | 4.5838 | 0.0001 |
| @CUMDN(LOG(GLOBAL_EXPORTS(−1))) | 1.0240 | 0.4966 | 2.0621 | 0.0463 |
| C | 24.6489 | 3.3695 | 7.3153 | 0.0000 |
| R-squared | 0.8773 | | | |
| Adjusted R-squared | 0.8363 | | | |
| S.E. of regression | 0.0213 | | | |
| Sum squared resid | 0.0136 | | | |
| Log likelihood | 106.1229 | | | |
| F-statistic | 21.4411 | | | |
| Prob(F-statistic) | 0.0000 | | | |

**Table A4.** Dynamic and long-run estimates (NARDL)—USD Claims.

| Short-run effects | | | |
|---|---|---|---|
| Variable | Coefficient | Std. Error | t-Statistic | Prob. |
| DLOG(GLOBAL_EXPORTS) | 0.0723 | 0.0086 | 8.3751 | 0.0000 |
| DLOG(GLOBAL_EXPORTS(-1)) | 0.0180 | 0.0078 | 2.3075 | 0.0292 |
| DLOG(GLOBAL_EXPORTS(-2)) | 0.0180 | 0.0086 | 2.0869 | 0.0468 |
| DLOG(GLOBAL_EXPORTS(-3)) | 0.0288 | 0.0093 | 3.0832 | 0.0048 |
| @DCUMDP(LOG(USD_CLAIMS)) | 0.0292 | 0.0054 | 5.3886 | 0.0000 |
| @DCUMDN(LOG(USD_CLAIMS)) | −0.0134 | 0.0052 | −2.5783 | 0.0159 |
| @DCUMDP(LOG(USD_CLAIMS(-1))) | 0.0035 | 0.0053 | 0.6662 | 0.5111 |
| @DCUMDN(LOG(USD_CLAIMS(-1))) | 0.0031 | 0.0052 | 0.5918 | 0.5591 |
| @DCUMDP(LOG(USD_CLAIMS(-2))) | 0.0051 | 0.0049 | 1.0477 | 0.3044 |
| @DCUMDN(LOG(USD_CLAIMS(-2))) | −0.0166 | 0.0049 | −3.3816 | 0.0023 |
| GFC | −0.0222 | 0.0039 | −5.6288 | 0.0000 |
| COVID | −0.0558 | 0.0052 | −10.7165 | 0.0000 |
| COINTEQ | −0.0467 | 0.0048 | −9.6766 | 0.0000 |
| Long-run effects | | | |
| LOG(WLD_EXPORTS(-1)) | −0.4754 | 0.5807 | −0.8187 | 0.4185 |
| @CUMDP(LOG(TOTAL_CLAIMS(-1))) | 0.2240 | 0.1542 | 1.4533 | 0.1551 |
| @CUMDN(LOG(TOTAL_CLAIMS(-1))) | −0.3239 | 0.2084 | −1.5545 | 0.1291 |
| C | 44.6130 | 16.5606 | 2.6939 | 0.0108 |
| R-squared | 0.9385 | | | |
| Adjusted R-squared | 0.9101 | | | |
| S.E. of regression | 0.0046 | | | |
| Sum squared resid | 0.0006 | | | |
| Log likelihood | 162.2276 | | | |
| F-statistic | 33.0663 | | | |
| Prob(F-statistic) | 0.0000 | | | |

## Note

[1] The same principles apply to the Euro, Yen, British Pound, etc., which can be collectively described as 'Eurocurrencies' (Battilossi et al. 2020).

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
