# Peer review of "Exploring the Dynamic Nexus between Cross-Border Dollar Claims and Global Economic Growth"

_economies, doi:10.3390/economies12030069_

Round 1
Reviewer 1 Report
Comments and Suggestions for Authors
Thanks for allowing me to review this interesting paper. A have a few comments on the topic, please see below.
1. The period 1980-2022 includes not just the Covid-19 and the GFC but also more important social events, such as the collapse of the Soviet Union and, most importantly, the fall of the Berlin wall. A dummy could be introduced for these events as well.
2. Global exports include or exclude US exports?
3. What happened in 1993 when cross border claims decreased?
4. Shouldn't the Euro-Dollar exchange rate also be included as it would significantly affect the deposit growth?
5. Not clear about the path of causation? Why wouldn't it be the other way around, i.e. that cross-border claims are affected by global GDP?
Comments on the Quality of English Language
The quality of English is good.
Author Response
Dear Reviewer
In the attached document we provide replies to your comments.
Thank you very much for the constructive comments that enabled us to improve our paper substantially.

Reviewer 2 Report
Comments and Suggestions for Authors
I think it is a good, well-structured paper that addresses a gap in the literature, raises research questions and evaluates them. That said, I think some changes should be made to improve the quality of the paper prior to publication.
1. This sentence “As money is a medium of exchange and does not directly create growth, it is safe to say that the availability of dollars ensured money supply was not a limiting factor in global growth and globalization in the post-war period, at least until 2007” seems to me a bit confusing.
2. I would insert a sentence linking the paragraph in which the lack of evidence on the relationship between Eurodollar and growth and the references to the two recent crises are mentioned. I think it would help the reader.
3. You are referring to research hypotheses you have not established.
4. Page 3. Lines 100-104. This paragraph would be a typical footnote, although I see you are not including anyone. So, perhaps I would trim it, and include somewhere that "the Eurodollar is not the only case, but it is the most important case".
5. “is, at best, non-existing”. I guess you want to put emphasis, but …
6. In any case, I would modify the title of section 3, since as you are saying this … is close literature at best 😊.
7. I wonder if any of the papers in the literature review use a similar methodology to yours. I believe you should say something about it.
8. Page 6. Last paragraph. You keep repeating it, are you sure there has been no attempt to cope with this problem?
9. You could have tested structural changes.
10. Some comments on the model:
a. Did you try filtering data? I mean, a linear tendency may not be the best option.
b. Independent Variables are selected ad hoc. You have to admit that an omitted variable problem can affect the results. Or maybe as long as you have data include standard variables related to economic growth.
c. You are writing an equation. My opinion here is that a reader not accustomed to using the methodology is likely to encounter problems linking your equation with Table 3. It would be more precise in section 4 to better understand section 5 and the reference to short-run, long-run….
d. Related to that, dummy variables should be explicitly included in equations.
e. Also related to that, now the note in Table 3 will be completely understood.
11. “are I(1)” is enough in the last sentence of section 4.
12. Change font size just before Table 4.
13. There are almost no comments on the results.
14. Concluding remarks. You wrote, in line with the introduction, “Our findings are of great significance for monetary authorities, emphasizing the need for a nuanced understanding and appreciation of the implications of the U.S. dollar's conducive role in shaping global dynamics and supporting economic growth on a world-wide scale, even in turbulent times.” You definitely should elaborate on it!
15. There are many “extra spaces” throughout the text, which makes the reading somewhat annoying.
Author Response
Dear Reviewer,
In the attached file we provide our replies to your comments.
We thank you for your constructive comments that enabled us to improve our paper significantly.

Round 2
Reviewer 2 Report
Comments and Suggestions for Authors
I am only somewhat happy with your answers. You should be more specific when answering in the check-list. In addition, some of my comments, especially those regarding the model and comments, were not properly answered. In any case, I guess the paper has improved to some extent and can be published.